# KIStego: Key-Independent Secure Image Distribution via Bipartite Structural Invariants

**Lijing Ren** [1]   **Denghui Zhang** [2]

## Abstract

Generative Image Steganography (GIS) embeds complex secrets within stego-images that are indistinguishable from the stochastic synthesis process itself. It achieves this by exploiting the reversible probability flow between Gaussian noise and the natural image manifold. However, existing steganography faces a key-dependency paradox: precise secret extraction usually requires an external private key or random seed to synchronize the denoising path. In this paper, we present KIStego, a training-free framework substituting cryptographic secrecy with structural redundancy for Key-Independent, high-resolution secure distribution. Our dual-guidance mechanism utilizes structural latent guidance to project secrets into a bipartite manifold via downsampling and halftoning, yielding self-synchronizing shares. Measurement posterior sampling leverages these shares as discrete invariants to steer a differentiable restoration. By backpropagating through a differentiable measurement surrogate, KIStego reconstructs high-fidelity continuous-tone details from sparse binary observations, mitigating fidelity loss from inversion drift. KIStego offers an endogenous secure image distribution paradigm by connecting discrete structural invariants with high-fidelity generative reconstruction.

## 1. Introduction

The proliferation of sensitive high-resolution visual data across decentralized digital ecosystems has catalyzed an urgent requirement for sophisticated covert communication mechanisms in untrusted networks. These strategies ensure

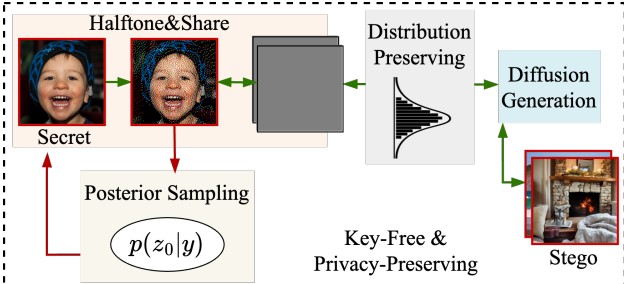

*Figure 1.* The pipeline of our Key-Independent secure image distribution scheme. The secret-sharing halftone image $y$ guides the diffusion sampling to recover the secret image $x_0$.

data privacy in critical domains such as healthcare imaging, biometric authentication, and remote sensing by concealing confidential information within seemingly innocuous media (Wen et al., 2023; Deng et al., 2023; Zhao et al., 2026). While traditional encryption techniques ensure confidentiality, they generate conspicuous noise-like patterns that alert adversaries to the existence of sensitive traffic, thereby stimulating cryptanalysis and interception in untrusted networks (Dong et al., 2021). Digital steganography has historically addressed this vulnerability by concealing payloads within digital containers. These modification-based paradigms fundamentally perturb the statistical distribution of the cover media (Filler et al., 2011; Ulutas et al., 2013; Yao et al., 2024), rendering them increasingly susceptible to advanced steganalysis detectors that exploit pixel-level or transform-domain artifacts to expose the presence of hidden signals (Aljarf et al., 2023).

By exploiting the reversible probability flow between Gaussian noise and the natural image manifold, generative image steganography (GIS) facilitates the synthesis of stego-images that remain statistically indistinguishable from genuine generative outputs, thereby embedding complex secrets within the stochastic synthesis process itself. The recent ubiquity of text-to-image AI-generated content (AIGC) has redefined the landscape, establishing denoising diffusion models as pervasive and inconspicuous stego-images for such covert communication (Yang et al., 2024c;b). Denoising diffusion models are now the leading steganography framework due to their high-fidelity mapping via reversible

[1]Institute of Artificial Intelligence, Guangdong Mechanical & Electrical Polytechnic, China [2]Cyberspace Institute of Advanced Technology, Guangzhou University, China. Correspondence to: Denghui Zhang <denghui.zhang@gzhu.edu.cn>.

*Proceedings of the 43$^{rd}$ International Conference on Machine Learning*, Seoul, South Korea. PMLR 306, 2026. Copyright 2026 by the author(s).

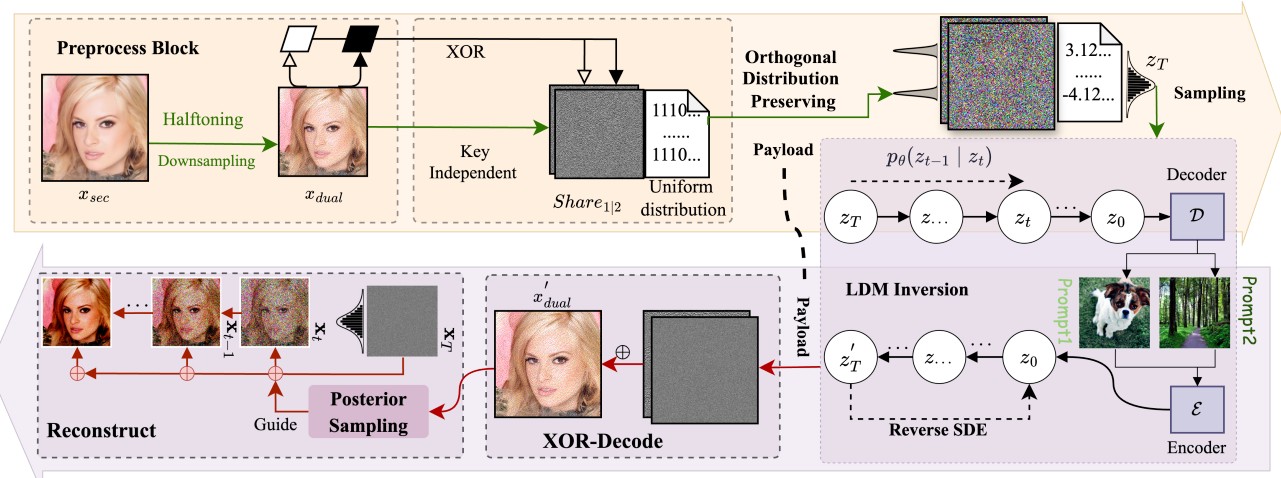

*Figure 2.* The proposed framework. Left: Secret preprocessing (downsampling + halftoning) converts high-resolution secret images into binary tensors, while posterior sampling guided by dual constraints reconstructs the secret from extracted bits. Middle: Orthogonal Gaussian mapping transforms binary secret tensors into Gaussian latent vectors matching the noise prior of diffusion models, enabling stealthy embedding. Right: ODE-based reversible sampling ensures structural decoupling between secret and stego-image, with text prompts controlling stego-image style.

probability flow (Ho et al., 2020; Chung et al., 2023; Peng et al., 2023; Zhou et al., 2025; Chung et al., 2023; Jiang et al., 2026). Early GIS operates directly in the pixel space using Denoising Diffusion Probabilistic Models (DDPMs) (Peng et al., 2023). Current paradigms have increasingly transitioned to Latent Diffusion Models (LDMs) to synthesize higher-quality and highly controllable stego-images (Rombach et al., 2022). However, because the generative process relies on a separate Variational Autoencoder (VAE), the steganographic mapping cannot be optimized in a fully integrated, end-to-end manner, thereby compounding the difficulty of precise secret recovery across the latent-to-pixel boundary. Modern diffusion-based steganographic methods, including CRoSS (Yu et al., 2023), DiffStega (Yang et al., 2024b), and Gaussian Shading (Yang et al., 2024c), exploit the probability flow Ordinary Differential Equation (ODE) to establish a deterministic trajectory between a Gaussian latent space and the image manifold. By projecting secret information into standard Gaussian noise, these methods synthesize stego-images that maintain near-perfect perceptual and statistical authenticity while theoretically allowing for high-capacity secret recovery. However, such diffusion-based steganography remains fundamentally tethered to a key-dependency paradox wherein both the sender and the receiver must pre-share an exogenous private key or seed to synchronize the stochastic denoising trajectory.

The reliance on out-of-band key distribution introduces logistical challenges and creates a critical single point of failure, as compromising a generative seed undermines the integrity of the entire channel (Thietke et al., 2025). While current research directions actively focus on stabilizing generation trajectories and refining noise-prior estimation to improve restoration quality, practical deployment and configurability constraints often force these systems to maintain a strict dependency on pre-shared keys. For instance, ODE-based methods like CRoSS remain sensitive to trajectory instability, where cumulative discretization errors lead to inversion drift that hinders high-fidelity signal recovery. Despite various steganographic refinement mechanisms proposed to address recovery issues (Kombrink et al., 2024), the reliance on out-of-band key distribution remains an unresolved practical bottleneck, highlights the need for a keyless transmission paradigm. Without an endogenous mechanism to distinguish message-carrying latents from infinite generative noise, the recovery process remains largely ill-posed unless assisted by pre-shared structural invariants (Chen et al., 2022; Wallace et al., 2023; Wang et al., 2024a; Ren & Zhang, 2026).

To address these critical limitations, we propose KIStego, a training-free framework that formulates secure image distribution as a self-synchronizing inverse problem on a pre-trained diffusion manifold $\mathcal{M}$. As shown in Figure 1, KIStego shifts the security paradigm from traditional external cryptographic shielding to endogenous security grounded in information-theoretic structural redundancy. While the process of embedding secrets into a carrier inevitably exposes the stego-image to external channel analysis and potential degradation, existing methods attempt to mitigate these vulnerabilities via fragile, key-dependent trajectory synchronization. In contrast, KIStego internalizes its defense mechanism within the structural representation of the carrier itself. By substituting exogenous cryptographic keys with information-theoretic structural invariants, our approach facilitates robust, high-fidelity reconstruction with-

out the necessity of out-of-band key management. This design enables endogenous distinguishability at the receiver side via internal mathematical logic, making the recovery process highly robust against trajectory drift and channel degradation without requiring private parameters.

Our primary contributions are summarized as follows:

- We introduce an endogenous security framework that replaces traditional, vulnerable exogenous cryptographic keys or private seeds with structural bipartite partitioning. By deriving security from the geometric properties of the image manifold, we project the secret payload into mathematically secure, uniformly distributed pixel shares governed by a public protocol matrix rather than a hidden private parameter. The intrinsic shares guarantee information-theoretic secrecy and enable reliable self-synchronization directly from the received stego-images, thereby mitigating the logistical overhead and single-point-of-failure vulnerabilities associated with out-of-band key distribution.

- We propose a unified framework that integrates structural latent guidance for distribution-preserving embedding with score-guided posterior sampling. Our architecture leverages the high-fidelity, highly controllable synthesis capabilities of latent representations. Our dual-guidance mechanism ensures that the synthesized stego-images adhere strictly to image prior.

- We formulate secret reconstruction as a constrained inverse problem, establishing a gradient-guided restoration engine that operates without task-specific model fine-tuning. By backpropagating through a differentiable measurement surrogate, we mitigate the trajectory-drift errors common in deterministic inversion.

## 2. Related Work

**Cover-based Steganography.** Traditional cover-based steganography facilitates data hiding by modifying the spatial or transform domains of a stego-image through algorithms such as STC encoding or wavelet-domain embedding (Filler et al., 2011; Ulutas et al., 2013; Yao et al., 2024). Recent deep learning paradigms have introduced autonomous embedding strategies, including adversarial perturbations in HiDDeN (Zhu et al., 2018) and generative adversarial networks in SteganoGAN (Zhang et al., 2019) to optimize image fidelity. Furthermore, architectures like Baluja (Baluja, 2020) and HiNet (Jing et al., 2021) employ encoder-decoder frameworks or invertible neural networks (INNs) (Lan et al., 2023; Wang et al., 2024b) to establish mappings between secret images and covers. Despite their high capacity, all cover-based approaches fundamentally perturb the statisti-

cal distribution of the stego-images, rendering them inherently susceptible to modern steganalysis tools that exploit modification-induced artifacts.

**Coverless Steganography and Key Dependency.** Diffusion models have demonstrated significant progress in CoverLess Steganography (CIS). Chen et al. (Chen et al., 2022) propose a rejection sampling strategy that encodes secret messages into latent vectors conforming to a Gaussian distribution. CRoSS (Yu et al., 2023) leverages a text-to-image diffusion model alongside modern conditioning tools to achieve high controllability and robustness in image steganography, where the deterministic DDIM sampling process is driven by two distinct prompts, including a private key and a public key. The method requires the sender and receiver to pre-share a private key to ensure accurate secret extraction. DiffStega (Yang et al., 2024b) operates by anchoring the steganographic mapping to pre-determined passwords and reference images, while incorporating Real-ESRGAN (Wang et al., 2021) as an external nonlinear enhancement tool to restore fidelity under various degradations. Gaussian Shading (Yang et al., 2024c) is predicated on a shared stream key, taking the ChaCha20 encryption algorithm to generate a random bitstream watermark. This bitstream is subsequently mapped to the latent space $x_T$ via distribution-preserving sampling, ensuring the final generation remains identical to that of a standard diffusion model. StegaDDPM (Peng et al., 2023) enables precise message extraction by requiring both parties to pre-share two private seeds that synchronize the Markov chain in the reverse diffusion process, where $Seed_1$ determines the starting point $x_T$ and $Seed_2$ dictates the stochastic transitions $\mathbf{x}_t$ at each step. While this guarantees high fidelity, it introduces the challenge of secure key distribution and management. These methods share a fundamental limitation in their inescapable dependence on pre-arranged keys for synchronization or encryption, underscoring the critical need for a more flexible Key-Independent GIS.

## 3. Methodology

### 3.1. Overview

As shown in Figure 2, our framework implements a dual-guidance mechanism that unifies distribution-preserving embedding with measurement-conditioned posterior sampling. Instead of relying on out-of-band key coordination or private random seeds to safeguard and align stochastic denoising paths, KIStego generates secure, uniformly distributed pixel shares that are mapped into the Gaussian-distributed latent space of an LDM. By employing structural bipartite partitioning, we project the high-dimensional secret into a statistically neutral manifold where the payload is decomposed into self-synchronizing shares. During recovery, the engine bridges the gap between these discrete structural

invariants and continuous-tone pixel reconstruction by back-propagating gradients through a differentiable surrogate of the measurement operator. This guidance mechanism steers the reverse sampling process, allowing the pre-trained LDM to restore high-fidelity details and mitigate inversion drift even under the empirical reconstruction constraints imposed by the LDM's VAE.

## 3.2. Probability Flow and the Diffusion Manifold

Let $\mathcal{X} \subset \mathbb{R}^d$ represent the high-dimensional image manifold and $\mathcal{Z} \cong \mathbb{R}^n$ denote the standard Gaussian latent space. A pre-trained diffusion model defines a bijective mapping $\Psi : \mathcal{X} \rightarrow \mathcal{Z}$ through the numerical integration of the probability flow ODE. The deterministic trajectory of the latent $\mathbf{x}_t$ is governed by the following SDE-derived ODE:

$$\frac{d\mathbf{x}_t}{dt} = f(t)\mathbf{x}_t - \frac{g^2(t)}{2}\nabla_{\mathbf{x}}\log p_t(\mathbf{x}_t), \quad (1)$$

where $f(t)$ and $g(t)$ are the drift and diffusion coefficients respectively, and $\nabla_{\mathbf{x}}\log p_t(\mathbf{x}_t)$ is the score function approximated by a time-dependent score network $\mathbf{s}_\theta(\mathbf{x}_t, t) \approx \nabla_{\mathbf{x}}\log p_t(\mathbf{x}_t)$. In the steganographic context, we treat the standard Gaussian latent $\mathbf{z}_T \in \mathcal{Z}$ at $t = T$ as the stego-images of the secret payload. The generative mapping $\Psi^{-1} : \mathcal{Z} \rightarrow \mathcal{X}$ synthesizes the stego-image $\mathbf{I}_{\text{stego}}$ while the encoding mapping $\Psi : \mathcal{X} \rightarrow \mathcal{Z}$ facilitates extraction.

## 3.3. Structural Invariant Projection via Discrete Measurement

To facilitate robust distinguishability without out-of-band trajectory information, we define a structural measurement operator $\mathcal{M} : \mathbb{R}^{H \times W \times C} \rightarrow \mathbb{F}_2^k$. For a high-resolution secret $\mathbf{I}_{\text{sec}}$, the operator is defined as a composition of a spatial pooling kernel $\mathcal{P}$ and a halftoning quantizer $\mathcal{Q}$:

$$\mathbf{B} = \mathcal{M}(\mathbf{I}_{\text{sec}}) = \mathcal{Q}(\mathcal{P}(\mathbf{I}_{\text{sec}})). \quad (2)$$

The pooling operator $\mathcal{P}$ reduces the spatial resolution of the $512 \times 512$ secret to align with the latent dimensions of the diffusion backbone. The quantization operator $\mathcal{Q}$ preserves the geometric topology and local density of the original signal within a binary manifold. By mapping continuous pixel intensities into discrete binary invariants, the embedded signal becomes significantly more resilient to the discretization errors—commonly known as inversion drift.

## 3.4. Bipartite Information-Theoretic Partitioning

To circumvent the problem inherent in Key-Independent noise sampling, we introduce a bipartite partitioning of the structural invariant $\mathbf{B} \in \mathbb{F}_2^k$ (Yan et al., 2019). We decompose $\mathbf{B}$ into two statistically neutral shares $\{\mathbf{S}_1, \mathbf{S}_2\}$ residing in the Galois field $\mathbb{F}_2$. The shares are generated such that:

$$\mathbf{S}_1 \sim \mathcal{U}(\mathbb{F}_2^k), \quad \mathbf{S}_2 = \mathbf{S}_1 \oplus \mathbf{B}, \quad (3)$$

where $\oplus$ denotes the bitwise XOR operation. This decomposition ensures that the distinguishability of the payload is endogenous to the stego-pair through the reconstruction logic $\mathbf{B} = \mathbf{S}_1 \oplus \mathbf{S}_2$. The individual shares individually reveal zero information regarding $\mathbf{B}$, facilitating secure transmission across separate untrusted channels.

## 3.5. Orthogonal Distribution-Preserving Latent Mapping

To ensure the stego-latents $\mathbf{z}_T^{(i)}$ adhere to the standard Gaussian prior $p(\mathbf{z}) = \mathcal{N}(0, \mathbf{I})$, we establish an orthogonal mapping $\mathbf{Q} \in \mathbb{O}(n)$. We standardize the flattened share $\mathbf{S}_i$ to a zero-mean, unit-variance vector $\tilde{\mathbf{v}}_i$ and apply the orthogonal transformation:

$$\mathbf{z}_T^{(i)} = \mathbf{Q}\tilde{\mathbf{v}}_i. \quad (4)$$

Since $\mathbf{Q}^\top\mathbf{Q} = \mathbf{I}$, the transformation preserves the spherical symmetry of the Gaussian manifold. This ensures that the generated latents $\mathbf{z}_T^{(i)}$ reside on the high-probability shell of the latent space, making $\mathbf{I}_{\text{stego}}^{(i)} = \Psi^{-1}(\mathbf{z}_T^{(i)})$ statistically indistinguishable from images synthesized via natural i.i.d. noise sampling.

## 3.6. Security and Statistical Undetectability

The security of KIStego is fundamentally grounded in information-theoretic secrecy and the structural invariance of the diffusion latent space.

**Information-Theoretic Perfect Secrecy.** Secrecy is established by the threshold property of the bipartite partitioning mechanism, which ensures that an adversary gains zero mutual information regarding the structural invariant from an individual stego-channel.

**Theorem 3.1** (Perfect Secrecy of Structural Shares). *For a discrete structural invariant $\mathbf{B} \in \mathbb{F}_2^k$, let $\{\mathbf{S}_1, \mathbf{S}_2\}$ be the bipartite shares residing in the Galois field $\mathbb{F}_2^k$. Each individual share $\mathbf{S}_i$ satisfies the condition of Shannon perfect secrecy such that $I(\mathbf{S}_i; \mathbf{B}) = 0$ for $i \in \{1, 2\}$.*

*Proof.* By construction, $\mathbf{S}_1$ is sampled from the uniform distribution $\mathcal{U}(\mathbb{F}_2^k)$, achieving maximal entropy $H(\mathbf{S}_1) = k$. The second share is derived via the deterministic mapping $\mathbf{S}_2 = \mathbf{S}_1 \oplus \mathbf{B}$. For any fixed invariant $\mathbf{B}$, the operator $f_{\mathbf{B}}(\mathbf{S}_1) = \mathbf{S}_1 \oplus \mathbf{B}$ defines a bijection on $\mathbb{F}_2^k$. Consequently, the conditional probability $p(\mathbf{S}_2|\mathbf{B})$ is equivalent to the marginal probability $p(\mathbf{S}_1)$, yielding $p(\mathbf{S}_2|\mathbf{B}) = 2^{-k}$. Since $p(\mathbf{S}_2) = \sum_{\mathbf{B}} p(\mathbf{S}_2|\mathbf{B})p(\mathbf{B}) = 2^{-k}\sum_{\mathbf{B}} p(\mathbf{B}) = 2^{-k}$, the joint distribution factorizes as $p(\mathbf{S}_i, \mathbf{B}) = p(\mathbf{S}_i)p(\mathbf{B})$. This factorization implies the mu-

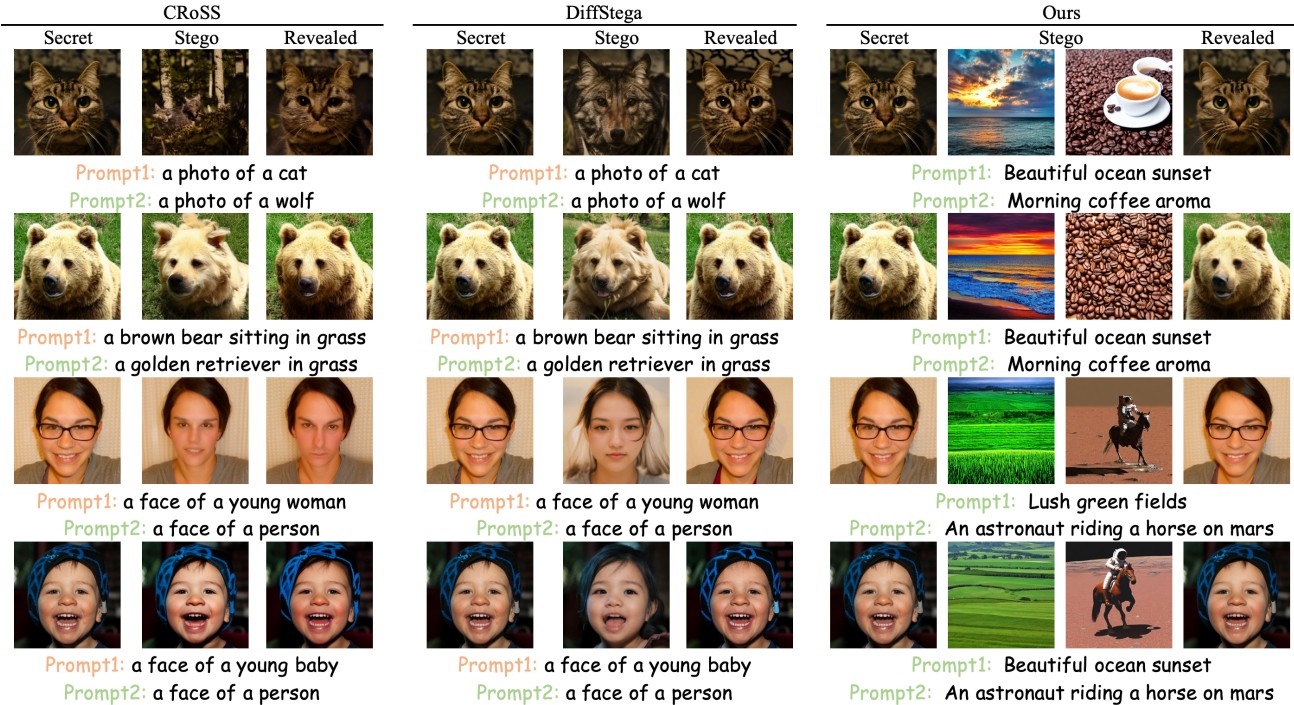

*Figure 3.* Visual comparison on UniStega. While CRoSS and DiffStega rely on local manifold perturbations, KIStego enables content-agnostic distribution. Our method hides high-fidelity secrets within entirely unrelated semantic priors without structural ghosting. The restoration shows high-frequency textural details from discrete structural measurements despite drastic distribution shifts.

tual information $I(\mathbf{S}_i; \mathbf{B}) = H(\mathbf{S}_i) - H(\mathbf{S}_i|\mathbf{B}) = 0$, proving that a single share reveals zero statistical information regarding the payload. $\square$

This model ensures that secret reconstruction is only possible through the endogenous XOR-logic $\mathbf{B} = \mathbf{S}_1 \oplus \mathbf{S}_2$, establishing a security framework that is independent of computational assumptions or exogenous key management.

**Statistical Invisibility in the Latent Manifold.** To prevent detection by steganalysis tools, the stego-latents $\mathbf{z}_T^{(i)}$ must be statistically indistinguishable from the standard Gaussian noise typically utilized in natural image synthesis.

**Theorem 3.2** (Rotational Invariance of Stego-Latents)**.** *The distribution of the stego-latent $\mathbf{z}_T^{(i)}$ under an orthogonal mapping $\mathbf{Q} \in \mathbb{O}(n)$ is invariant and converges asymptotically to the standard Gaussian prior $\mathcal{N}(0, \mathbf{I})$ as the latent dimension $n \to \infty$.*

*Proof.* The standard Gaussian density $p(\mathbf{z}) \propto \exp(-0.5\|\mathbf{z}\|_2^2)$ exhibits spherical symmetry and is invariant under any orthogonal transformation $\mathbf{Q}$. Let $\tilde{\mathbf{v}}_i \in \mathbb{R}^n$ be the standardized vector derived from share $\mathbf{S}_i$, characterized by i.i.d. elements with zero mean and unit variance. According to the multivariate Central Limit Theorem, the distribution of $\tilde{\mathbf{v}}_i$ converges in distribution to $\mathcal{N}(0, \mathbf{I})$ as $n$ increases. The covariance of the mapped latent $\mathbf{z}_T^{(i)} = \mathbf{Q}\tilde{\mathbf{v}}_i$ is given by:

$$\begin{aligned} \mathrm{Cov}(\mathbf{z}_T^{(i)}) = \mathbb{E}[(\mathbf{Q}\tilde{\mathbf{v}}_i)(\mathbf{Q}\tilde{\mathbf{v}}_i)^\top] &= \mathbf{Q}\mathbb{E}[\tilde{\mathbf{v}}_i \tilde{\mathbf{v}}_i^\top]\mathbf{Q}^\top \\ &= \mathbf{Q}\mathbf{I}\mathbf{Q}^\top = \mathbf{I}. \end{aligned} \tag{5}$$

Since both the mean and covariance match the generative prior, the resulting stego-latent occupies the high-probability density regions of the manifold, remaining statistically indistinguishable from the noise manifold utilized in unconditional diffusion sampling. $\square$

**Endogenous Synchronization and Key-Independent Security.** KIStego departs from seed-dependent synchronization where security is tethered to the secrecy of a random seed $s \in \mathcal{S}$. In our framework, secrecy is grounded in the information-theoretic gap between bipartite shares rather than the secrecy of the probability flow trajectory. Even if the orthogonal matrix $\mathbf{Q}$ or the generative seeds are publicly known, an adversary cannot recover $\mathbf{B}$ without access to both communication channels. This shift from cryptographic secrecy to structural secrecy enables self-synchronizing extraction, where the receiver utilizes structural invariants of the synthesized pixels to drive the score-guided posterior restoration. Reliability is thus governed by endogenous bipartite logic rather than the exogenous synchronization of stochastic trajectories, resolving the distinguishability problem in Key-Independent generative steganography.

| Method | Concealment Efficacy (Stego vs. Secret) | | | | Recovery Fidelity | | | | Generative Quality | |
|---|---|---|---|---|---|---|---|---|---|---|
| | PSNR↓ | SSIM↓ | LPIPS↑ | ID Sim↓ | PSNR↑ | SSIM↑ | LPIPS↓ | ID Sim↑ | CLIP Score↑ | NIQE↓ |
| CRoSS | 19.5286 | 0.6002 | 0.3734 | 0.4621 | 21.3026 | 0.6559 | 0.2474 | 0.6442 | 0.2372 | 4.1985 |
| DiffStega | 18.8524 | 0.5847 | 0.4106 | 0.2255 | 23.9374 | 0.7065 | 0.1633 | 0.7797 | 0.2524 | 3.7659 |
| **KIStego (Ours)** | **9.5065** | **0.1979** | **0.7970** | **0.0106** | **24.2491** | **0.7280** | **0.1610** | **0.8249** | **0.2941** | **3.2480** |

*Table 1.* Quantitative validation of manifold isolation and posterior convergence. The results establish a new Pareto frontier in generative steganography, where KIStego achieves near-optimal statistical uncoupling between the secret payload and the stego-images. Unlike deterministic inversion paradigms (CRoSS, DiffStega) that suffer from semantic leakage and identity blurring, our framework demonstrates that grounding recovery in structural invariants reconciles payload capacity, statistical invisibility, and reconstruction fidelity. The superior CLIP and NIQE scores confirm that our orthogonal mapping preserves the integrity of the generative prior, ensuring that the stego-distribution is indistinguishable from the natural image manifold.

## 3.7. Score-Guided Posterior Restoration via Differentiable Surrogates

The final phase of the KIStego pipeline involves reconstructing high-fidelity continuous-tone pixels $\mathbf{x}_0$ from the XOR-recovered structural invariant $\mathbf{B}$ (hereafter denoted as the observation $\mathbf{y} \in \{0,1\}^k$ for clarity) (Zirvi et al., 2025). We reformulate this recovery as a constrained inverse problem, leveraging the generative prior of the diffusion model to hallucinate details filtered during the discrete projection.

**Degradation Model.** To recover the original secret $\mathbf{x}_0$ from the reconstructed halftone observation $\mathbf{y}$, we design a training-free posterior sampling method. This addresses the ill-posed nature of inverse halftoning, which arises from the severe information loss and the non-linearity inherent in the binarization process. Since our generative backbone utilizes a latent diffusion models, the sampling process occurs within a compressed manifold $\mathcal{Z}$ where the clean secret $\mathbf{x}_0$ corresponds to a latent vector $\mathbf{z}_0$ such that $\mathbf{x}_0 = \mathcal{D}(\mathbf{z}_0)$. Traditional inverse methods often fail in this regime due to the non-convex optimization landscape and the lack of strong structural priors. We formulate the halftoning process as a noisy nonlinear degradation on the decoded space:

$$\mathbf{y} = \mathcal{A}(\mathcal{D}(\mathbf{z}_0)) + \eta, \quad \eta \sim \mathcal{N}(0, \sigma^2 \mathbf{I}), \qquad (6)$$

where $\mathcal{A}(\cdot)$ represents the non-differentiable halftoning operator and $\eta$ captures the acquisition noise introduced during quantization and potential channel transmission.

**Posterior Sampling via Differentiable Guidance.** The restoration task is to sample from the posterior distribution $p(\mathbf{z}_0|\mathbf{y}) \propto p(\mathbf{y}|\mathbf{z}_0)p(\mathbf{z}_0)$, where $p(\mathbf{z}_0)$ is the manifold prior learned by the latent diffusion backbone. In the score-based framework, the transition from $\mathbf{z}_t$ to $\mathbf{z}_{t-1}$ is steered by the measurement consistency in the pixel domain. However, since the gradient $\nabla_{\mathbf{z}_t} \log p(\mathbf{y}|\mathbf{z}_t)$ is intractable due to the discontinuity of $\mathcal{A}(\cdot)$, we introduce a differentiable surrogate $\mathcal{M}_\tau$ acting on decoded images:

$$\mathcal{M}_\tau(\mathbf{x}) = \text{Sigmoid}\left(\tau(\mathcal{P}(\mathbf{x}) - \mathbf{d})\right), \qquad (7)$$

where $\mathcal{P}(\cdot)$ is the pooling operator, $\mathbf{d}$ is the dithering matrix,

and $\tau$ is a sharpness parameter. The latent reconstruction trajectory is computed via:

$$\mathbf{z}_{t-1} = \Psi_{\text{step}}(\mathbf{z}_t, t) - \zeta_t \nabla_{\mathbf{z}_t} \|\mathbf{y} - \mathcal{M}_\tau(\mathcal{D}(\hat{\mathbf{z}}_0(\mathbf{z}_t)))\|_2^2. \quad (8)$$

Here, $\hat{\mathbf{z}}_0(\mathbf{z}_t)$ constitutes the Tweedie estimate of the clean latent at timestep $t$:

$$\hat{\mathbf{z}}_0(\mathbf{z}_t) = \frac{\mathbf{z}_t - \sqrt{1 - \bar{\alpha}_t}\mathbf{s}_\theta(\mathbf{z}_t, t)}{\sqrt{\bar{\alpha}_t}}, \qquad (9)$$

where $\mathbf{s}_\theta(\mathbf{z}_t, t)$ is the predicted score in the latent space. This gradient-based guidance ensures that the high-resolution pixel output $\mathcal{D}(\mathbf{z}_0)$ adheres to the structural invariants captured in the binary shares, while the diffusion prior restores high-frequency textural components and continuous color gradients. By employing this differentiable posterior sampling, our framework bridges the gap between discrete structural invariants and high-fidelity pixel reconstruction, ensuring robust recovery even when the latent trajectory is subject to ODE discretization drift.

## 4. Experiments

### 4.1. Experimental Setup

**Implementation Details.** Our framework is implemented using the PyTorch library and the HuggingFace Diffusers pipeline (Jain, 2022). All experiments are executed on an Ubuntu workstation equipped with an NVIDIA RTX 4090 GPU (24GB). We evaluate the task of concealing a full-size secret image $\mathbf{I}_{\text{sec}}$ within a stego-image $\mathbf{I}_{\text{stego}}$ of identical dimensions ($512 \times 512$). We focus on assessing its performance in high-resolution cross-domain steganography, highlighting its superiority in reconstruction fidelity, generative naturalness, and Key-Independent reliability compared to state-of-the-art diffusion-based baselines. We utilize the pre-trained Stable Diffusion v1.5 as the generative backbone, which defines the latent manifold $\mathcal{Z}$ with a spatial compression factor of $f = 8$ via its VAE component. For the measurement-conditioned posterior sampling, the consistency guidance scale $\lambda$ and the differentiable surrogate

sharpness parameter $\tau$ are empirically set to 5 and 10, respectively, to maintain an optimal balance.

**Datasets and Scenarios.** We evaluate our framework on the UniStega benchmark (Yang et al., 2024b), comprising 100 image-prompt pairs curated from MS-COCO, AFHQ, FFHQ, and CelebA-HQ at $512^2$ resolution. The suite spans three representative scenarios: content variation, style transfer, and semantic similarity. We follow the standard UniStega prompts and protocols to maintain a fair comparison.

**Evaluation Metrics.** We quantify the performance of our framework through hierarchical dimensions. We measure the pixel-level and perceptual fidelity of the recovered secret $\hat{\mathbf{I}}_{sec}$ with PSNR, SSIM, and LPIPS (Detlefsen et al., 2022). For face-specific concealment, we employ the ID cosine similarity calculated via a pre-trained FaceNet to evaluate the preservation of identity-critical features. To ensure that the embedding process does not distort the model's learned distribution, we evaluate the stego-images $\mathbf{I}_{stego}$ at the population level using the CLIP Score to assess spatio-semantic consistency with the input prompts. We utilize the NIQE (Mittal et al., 2012) as a no-reference perceptual metric, where lower values signify higher visual plausibility and fewer generative artifacts. The reported concealment efficacy represents the mean across the synthesized stego-pairs.

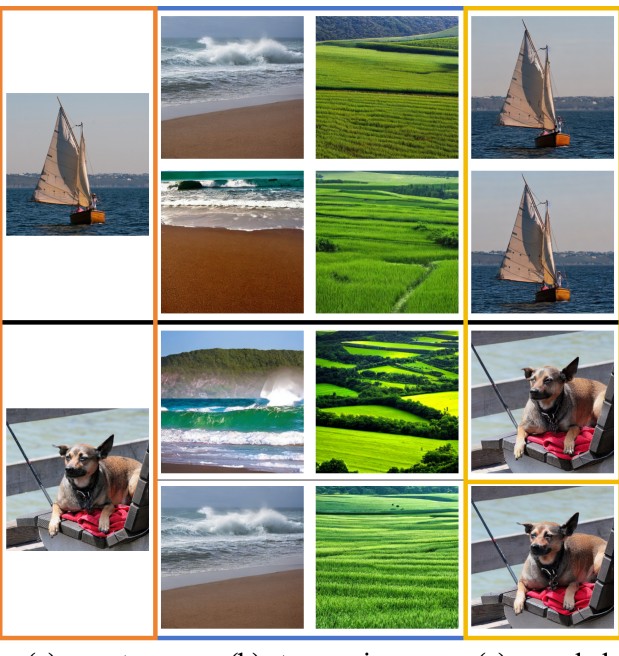

(a) secret      (b) stego-pairs      (c) revealed

*Figure 4.* Our cryptographic steganography realizes one-time pad security by producing diverse stego-pairs and ensuring secrecy.

### 4.2. Main Results

**Quantitative Performance.** The empirical results in Table 1 establish high-quality GIS by reconciling high-capacity dis-

tribution with statistical stealth. Our framework demonstrates superior concealment efficacy where the stego-images are uncoupled from the secret payload. While baselines like CRoSS exhibit semantic leakage characterized by residual identity cues within the stego-image, KIStego achieves near-perfect manifold isolation. The marginal gains suggest that our orthogonal latent mapping preserves the integrity of the generative prior better than modification-based latent embedding, ensuring the stego-distribution remains indistinguishable from the natural image manifold.

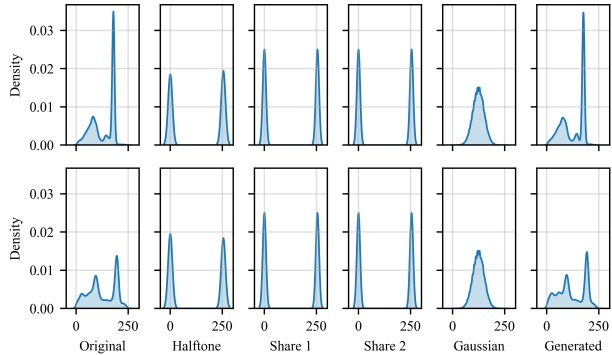

*Figure 5.* Security analysis based on statistical characteristics of histograms for Figure 4. The original sensitive data is on the left. The other columns show the encrypted pixel distribution, the corresponding decrypted data, and the transformed Gaussian latent and generative distribution.

**Qualitative Analysis.** Figure 3 illustrates a paradigm shift from semantic proximity to complete manifold decoupling. Baselines like CRoSS and DiffStega rely on deterministic ODE inversion, necessitating structural commonalities between payloads and stego-images (e.g., Cat-to-Wolf), which frequently induces keys structural leakage or spectral ghosting. Conversely, KIStego achieves complete semantic orthogonality, severing payload-stego ties. By projecting bipartite shares into a distribution-preserving latent space, our structural latent guidance enables arbitrary stego-images synthesis guided solely by stego-prompts. This decoupling does not compromise precision. Our measurement-conditioned posterior sampling generates high-fidelity details when the generative prior is semantically unrelated to the secret. The synthesized stego-images look pristine and artifact-free, indistinguishable from natural generative outputs. Grounding recovery in discrete invariants rather than reversing trajectories lets our method infer high-fidelity details even with a semantically orthogonal prior to the secret. This semantic independence preserves reconstruction accuracy. Baselines struggle with distribution shifts. While our measurement-conditioned posterior sampling maintains stable restoration across large semantic gaps.

**Security of Shares.** The statistical integrity of our bipartite structural shares is evaluated through a comprehensive

*Table 2.* PSNR for restoration robustness under manifold perturbations. KIStego exhibits superior resilience compared to deterministic inversion baselines like CRoSS and DiffStega. Our framework preserves structural integrity through measurement invariants across a range of degradations, including Gaussian blur ($7 \times 7$, $\sigma = 10$), Gaussian noise ($\sigma = 0.1$), JPEG compression ($Q = 70$), and spatial scaling ($0.5\times$).

| Method | Clean | Blur | Noise | JPEG | Scaling |
|---|---|---|---|---|---|
| HiNet | **46.15** | 10.26 | 11.45 | 10.95 | 9.84 |
| CRoSS | 21.25 | 20.08 | 19.35 | 20.19 | 18.27 |
| DiffStega | 23.94 | 20.85 | 18.62 | 21.16 | 19.45 |
| **Ours** | 24.25 | **22.48** | **21.92** | **23.14** | **21.56** |

*Table 3.* Steganalysis resistance on the UniStega dataset. We report the detection accuracy ($P_d$) for SRNet, YeNet, and SiaStegNet detectors. A value of 0.50 indicates ideal statistical invisibility.

| Detector | Ours | DiffStega | CRoSS | HiNet |
|---|---|---|---|---|
| SRNet | **0.502** | 0.546 | 0.582 | 0.765 |
| YeNet | **0.501** | 0.538 | 0.565 | 0.724 |
| SiaStegNet | **0.503** | 0.551 | 0.594 | 0.812 |

*Table 4.* Ablation of hiding capacity and resolution scaling. We analyze the trade-off between Bit-Per-Pixel (BPP) and recovery fidelity on a $512 \times 512$ stego-image. The $128^2$ resolution (Res.) defines our latent baseline, achieving a 1:1 bit-to-dimension alignment with the VAE bottleneck ($64 \times 64 \times 4$). Higher BPP are achieved through spatial multiplexing of multiple bipartite shares.

| Res. | Shares | BPP | PSNR ↑ | SSIM ↑ | LPIPS ↓ |
|---|---|---|---|---|---|
| $128 \times 128$ | 1 | 0.5 | 24.25 | 0.728 | 0.161 |
| $90 \times 90$ | 2 | 1.0 | 23.14 | 0.695 | 0.184 |
| $64 \times 64$ | 4 | 2.0 | 21.82 | 0.642 | 0.215 |
| $45 \times 45$ | 8 | 4.0 | 20.15 | 0.584 | 0.258 |

*Table 5.* Ablation study on differentiable measurement guidance and inverse halftoning strategies.

| Method | PSNR ↑ | SSIM ↑ | LPIPS ↓ |
|---|---|---|---|
| Latent-DPS | 18.56 | 0.781 | 0.285 |
| Latent-DPS + ED (STE) | 21.12 | 0.805 | 0.182 |
| Ours-HT | 20.85 | 0.824 | 0.245 |
| **Ours (Full)** | **24.25** | **0.858** | **0.161** |

histogram analysis as illustrated in Figure 4 and Figure 5, where the transition from complex semantic content to information-theoretic entropy is quantitatively confirmed. While the original secret image exhibits a multi-modal pixel distribution with significant fluctuations that characterize its high-resolution textures, the encrypted shares approach a state of perfect uniformity, which eliminates all discernible statistical patterns related to color and luminance. The alignment between these uniform structural shares and the standard Gaussian latent space through our orthogonal mapping prevents adversaries from distinguishing stego-latents from the model's inherent generative noise.

**Anti-Steganalysis.** The statistical indistinguishability of KIStego is quantitatively validated against detectors including SRNet, YeNet, and SiaStegNet (Boroumand et al., 2019; You et al., 2021), where it achieves a near-ideal detection accuracy ($P_d$) of approximately 0.50, establishing a stego-distribution that is virtually identical to the model's learned natural manifold. While deterministic inversion-based baselines like CRoSS and DiffStega often introduce detectable high-frequency residuals or structural ghosts during the ODE inversion process, our framework projects bipartite shares into the Gaussian latent space via an orthogonal transformation $\mathbf{Q}$. These standardized binary shares converge to the standard Gaussian prior $\mathcal{N}(0, \mathbf{I})$ in high-dimensional settings ($n = 512^2$). The rotational invariance of the Gaussian density ensures stego-images remain statistically indistinguishable from natural generative samples. Bipartite channel symmetry ensures symmetric protection, resolving the distinguishability problem by grounding stealth in manifold geometry rather than relying on a secret random generator.

**Robustness Analysis.** Table 2 shows a critical rupture between deterministic inversion paradigms and our measurement-conditioned restoration framework. While modification-based methods like HiNet collapse under manifold perturbations due to the systematic destruction of high-frequency secret coefficients by low-pass channel degradations, and inversion-based baselines such as CRoSS and DiffStega suffer from trajectory divergence when stego-latent coordinates shift, KIStego demonstrates superior resilience by anchoring the payload to discrete structural invariants. This binarization process acts as a non-linear geometric filter that preserves the topological integrity of the secret amidst severe textural blurring or JPEG-induced artifacts, ensuring that the geometric skeleton remains identifiable across adverse channels. By shifting the reconstruction paradigm from brittle trajectory reversal to score-guided posterior sampling, our framework leverages the generation potency of the diffusion prior to reconstruct high-fidelity continuous-tone details from corrupted observations.

### 4.3. Ablation Study

**Manifold Capacity and Bitstream Synchronization.** We can scale downsampling to surpass the 0.5 BPP baseline, achieving higher payload densities via multiplexing between the structural bitstream and the latent manifold capacity. Table 4 reveals a latent bottleneck baseline where structural invariants and latent dimensions achieve optimal parity. By systematically scaling the secret resolution, our framework facilitates spatial multiplexing to consolidate multiple bipartite measurements into a single stego-latent, thereby expanding the effective payload capacity without necessitating architectural recalibration. This rise in bit density inevitably

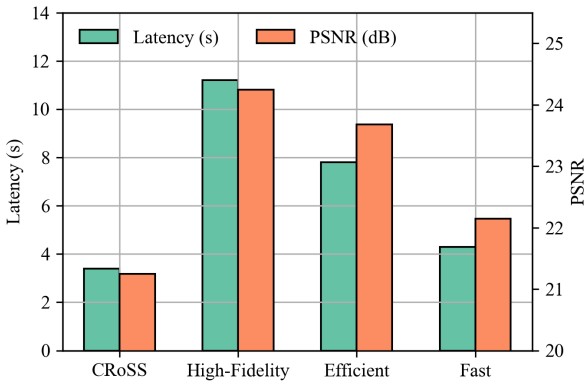

*Figure 6.* Comparative analysis of computational latency (inference time) and restoration fidelity (PSNR) across different configurations of KIStego, illustrating the trade-offs between the high-fidelity, efficient, and fast recovery modes.

causes progressive saturation, forcing the generative prior to contend with greater competition in the latent bottleneck to extract continuous-tone details from denser observations. The graceful degradation of restoration fidelity across these varying latent loads confirms that our score-guided posterior sampling bridges the gap between discrete structural invariants and high-fidelity pixel reconstruction.

**Efficacy of Differentiable Measurement Guidance.** The results in Table 5 establish the critical role of our differentiable halftoning-guided inverse sampling mechanism in reconciling discrete structural invariants with continuous-tone pixel reconstruction. The performance gap between the full framework and the Latent-DPS+error diffusion (ED) baseline (Song et al., 2024), highlighting the limitations of utilizing the Straight-Through Estimator (STE) (Xia et al., 2021) to approximate the non-differentiable (ED) operator. The crude gradient approximations introduce a catastrophic mismatch between the forward physical model and the backward sampling trajectory. This gradient misalignment precludes the diffusion process from navigating the non-convex optimization landscape associated with inverse halftoning. Furthermore, the failure of the unguided Latent-DPS framework confirms that the generative prior alone lacks sufficient directional guidance to converge toward the original image manifold in the absence of a precise structural constraint.

**Computational Efficiency.** While KIStego leverages DPS to achieve high-fidelity restoration, the iterative backpropagation process incurs a higher computational burden. To mitigate this, KIStego enables a trade-off between reconstruction speed and fidelity via three recovery modes in Figure 6. The high-fidelity mode prioritizes optimal quality at higher latency. The efficient mode applies measurement-guided gradients at decimated intervals (every 10 steps) (Yang et al., 2024a), reducing latency by approximately

30% while maintaining superior PSNR over baselines like CRoSS. The fast mode bypasses gradient-based guidance entirely, relying on direct latent projection of structural invariants for rapid recovery (Vaisman et al., 2025). This architectural flexibility allows KIStego to adapt seamlessly to diverse hardware constraints and security requirements.

## 5. Conclusion

In this paper, we present KIStego to resolve the key-dependency paradox in generative image distribution by grounding security in information-theoretic structural redundancy rather than exogenous cryptographic keys. We facilitate a self-synchronizing communication protocol where the recovery of high-resolution secrets is governed by the endogenous bipartite logic of the image manifold. Through the integration of structural latent guidance and orthogonal distribution-preserving mapping, we achieve semantic orthogonality between the secret payload and the stego-images. Extensive experiments show KIStego achieves high-fidelity and secure recovery.

## Impact Statement

This work utilizes established, publicly available benchmarks essential for ensuring reproducibility and fair performance comparison in the field. We acknowledge that the original collection of human-subject images within these standard datasets often lacked explicit individual consent. While our method is designed to be content-independent. These images are used strictly for the academic benchmarking of technical performance.

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

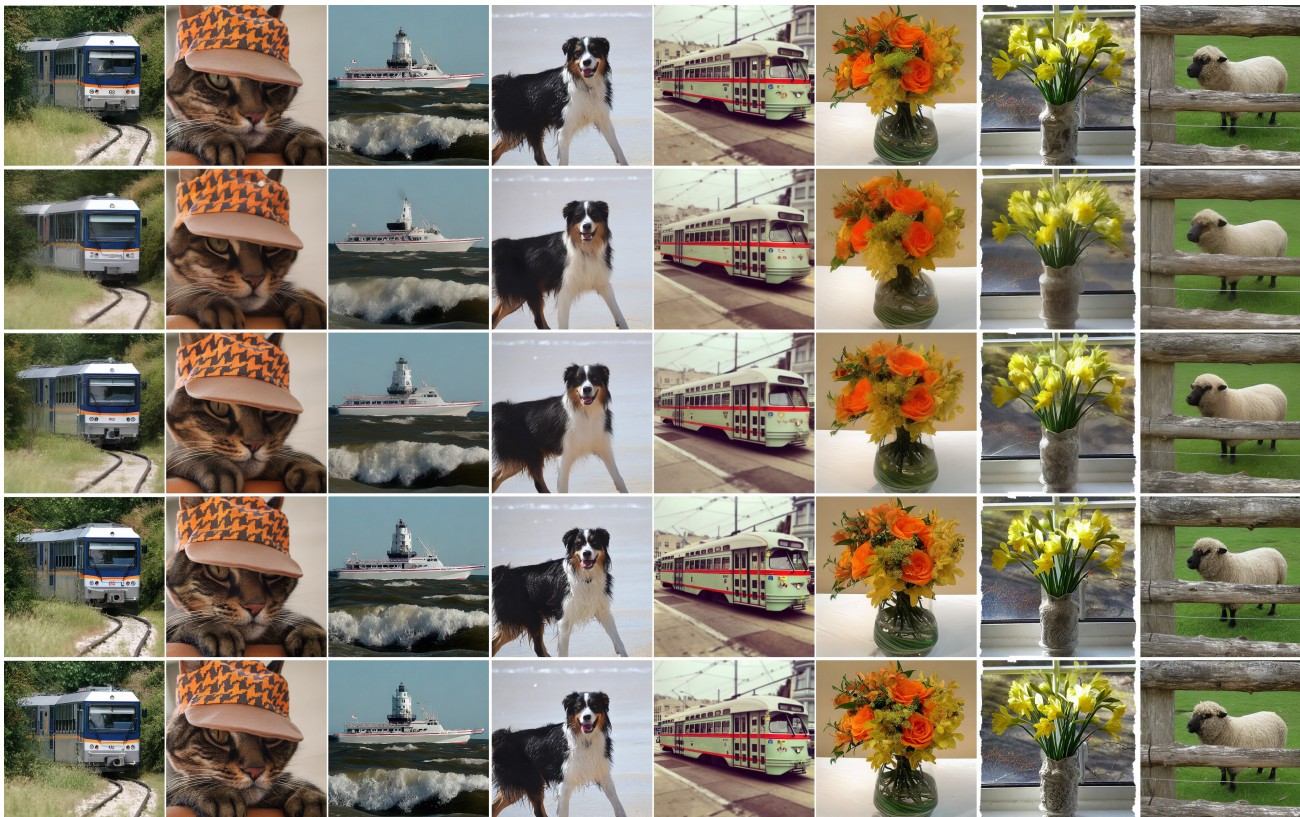

*Figure 7.* Qualitative comparison of reconstructed images across different surrogate sharpness parameters. The rows display results corresponding to increasing values of the sharpness parameter from top to bottom. The columns depict different instances.

## A. More Qualitative Results

In Figure 7, we present extended qualitative reconstructions across these parameter scales to visually substantiate the optimization trade-offs discussed above. At the lower end, relaxed gradient constraints produce a noticeable softening of high-frequency textures, especially in detailed regions such as animal fur and intricate flower petals. The intermediate range consistently yields the most visually balanced continuous-tone details, preserving sharp boundaries and natural textures without introducing artificial distortions. This smooth visual progression demonstrates that our framework does not require hand-crafted, image-specific configurations to maintain high-fidelity output across diverse content categories.

## B. Sensitivity Analysis

We employ $\beta$ in our differentiable measurement operator to balance gradient smoothness with approximation accuracy. As shown in Figure 8, quantitative metrics form a stable performance plateau across a wide range of scales, indicating that the optimization is robust and requires no per-image tuning. This stability arises from two opposing optimization boundaries. Under-scaling produces an overly relaxed approximation of the measurement boundary: while it aids initial gradient propagation, it lacks the precision to enforce discrete structural constraints. Conversely, over-scaling causes the surrogate to approximate a non-differentiable step function, resulting in severe gradient saturation and divergent posterior sampling.

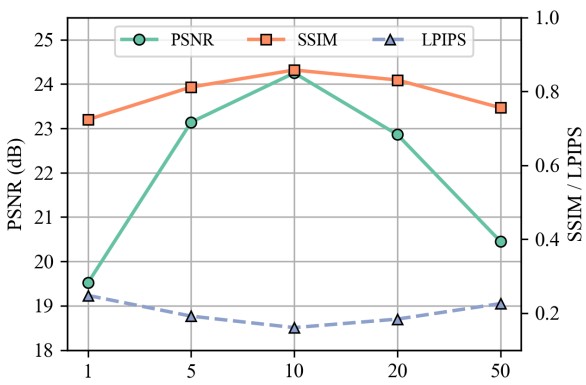

*Figure 8.* Sensitivity analysis of $\beta$ on restoration fidelity.

