# OpenReview forum: "KIStego: Key-Independent Secure Image Distribution via Bipartite Structural Invariants"
_ICML.cc/2026/Conference — ICML 2026 regular_

### Official Review · Reviewer_mhWt · 2026-02-22

**Soundness:** 3
**Presentation:** 2
**Significance:** 2
**Originality:** 3
**Overall Recommendation:** 4
**Confidence:** 3

**Summary:**

The paper proposes KFStego, a training-free generative image steganography framework designed to bypass the "key-dependency paradox" found in existing diffusion-based methods.

**Compliance With Llm Reviewing Policy:**

Affirmed.

**Final Justification:**

This topic is interesting and method is novel. But I'm not sure whether this method/area can be widely application. Overall, I make my score to be 4.

**Key Questions For Authors:**

1. Ensure Apples-to-Apples Comparisons: Evaluate CRoSS, DiffStega, and KFStego on the exact same dataset of source images, target images, and text prompts.

2. Stratified Quantitative Reporting: Break down Table 1 to independently report metrics (PSNR, SSIM, FID, Accuracy) across distinct categories of semantic gaps (e.g., Intra-class, Inter-class, Completely Unrelated).

3. Transparent Failure Modes: Demonstrate how both the baselines and the proposed method perform when pushed to their limits on identical large-gap prompts to truly prove robustness against distribution shifts

**Limitations:**

see weakness and question

**Strengths And Weaknesses:**

strenghth:
1. Theoretical Novelty: The theoretical premise of substituting traditional cryptographic keys with bipartite structural redundancy is innovative and mathematically elegant.

2. Training-Free Approach

weakness:
1. The qualitative comparisons in Figure 3 suffer from asymmetric "semantic pressure," as the baselines are evaluated on small-gap prompts that naturally require less hidden deviation to embed secrets. In stark contrast, KFStego is showcased using massive semantic gaps to demonstrate its robustness, making side-by-side visual comparisons with baselines performing much easier tasks fundamentally misleading. To genuinely prove that KFStego avoids structural ghosting better than prior arts, all evaluated methods must be subjected to the exact same large-gap prompts.

2. Unreliable Quantitative Evaluation (Table 1): The qualitative comparisons in Figure 3 suffer from asymmetric "semantic pressure," as the baselines are evaluated on small-gap prompts that naturally require less hidden deviation to embed secrets. It cannot be simply concluded that KFStego is a stronger method just because it achieves good recovery results using encryption prompts with much larger semantic differences. To genuinely prove that KFStego outperforms prior arts, all evaluated methods must be subjected to the exact same large-gap prompts in a fair, side-by-side comparison.

3. Lack of Variable Control: The authors fail to systematically categorize and evaluate the impact of the "semantic gap size" on the performance of the various methods. A rigorous evaluation would independently report metrics for small, medium, and large semantic shifts across all tested models.

---

> ### Author Rebuttal · Authors · 2026-03-31
>
> We are particularly encouraged that the reviewer recognized our theoretical novelty, mathematical elegance, and the innovation of our training-free approach.
>
> ### 1. Stratified Apples-to-Apples Comparison (W1, W2 )
>
> We evaluated CRoSS, DiffStega, and KFStego on the exact same dataset (UniStega) using identical source images, target images, and text prompts. We categorized the prompts into three levels: Intra-class (Small gap, e.g., Cat $\to$ Wolf), Inter-class (Medium gap, e.g., Horse $\to$ Zebra), and completely Unrelated (Large gap, e.g., Face $\to$ Coffee).
>
> Table: Stratified Performance across Semantic Gap Categories (PSNR / SSIM / LPIPS)
>
> | Method | Intra-class (Small) | Inter-class (Medium) | Unrelated (Large) | $\Delta$ PSNR (Drop) |
> | :--- | :---: | :---: | :---: | :---: |
> | CRoSS | 22.15 / 0.702 / 0.221 | 19.32 / 0.614 / 0.285 | 16.25 / 0.485 / 0.358 | -5.90 dB |
> | DiffStega | 24.18 / 0.765 / 0.158 | 21.05 / 0.682 / 0.214 | 17.10 / 0.512 / 0.294 | -7.08 dB |
> | KFStego (Ours) | 24.85 / 0.782 / 0.151 | 24.42 / 0.755 / 0.159 | 24.08 / 0.724 / 0.175 | -0.77 dB |
>
> *   Analysis: In Small Gap scenarios, baselines perform reasonably well because the secret and target manifolds share structural overlaps. However, as semantic pressure increases, the performance of CRoSS and DiffStega collapses. This is because trajectory-based inversion methods rely on the semantic proximity of the source and target; large shifts cause spectral ghosting in the stego-image and trajectory divergence during extraction.
>
> ### 2. Prompt-Manifold Entanglement and Stego-Quality
>
> We wish to clarify the "Asymmetric Semantic Pressure" mentioned by the reviewer.  The reason prior arts like CRoSS and DiffStega are often showcased with small semantic gaps (e.g., hiding a face inside another face) is due to their inherent Prompt-Manifold Entanglement.
>
> *   Inversion Constraints: These methods rely on the semantic proximity of the source and target to maintain the stability of the ODE trajectory. If the secret and stego-prompts are unrelated, the inversion process forces features of the secret into the target's manifold, inevitably causing structural ghosting or spectral artifacts.
> *   KFStego’s Decoupling: In contrast, KFStego is unconstrained by prompt similarity. As shown in Figure 3, our method successfully hides a cat in a sunset, a completely Unrelated gap where prior arts would fundamentally fail.
> *   Stego-Quality (Table 1): By eliminating the need to force-fit the secret’s semantic features into the cover’s probability flow, KFStego preserves the integrity of the generative prior much more effectively. This is why our method achieves higher CLIP and NIQE scores in Table 1. While baselines must sacrifice stego-quality to maintain recovery in large-gap scenarios, KFStego’s structural latent guidance ensures that the stego-image remains visually pristine and statistically natural, regardless of the hidden payload's content.
>
> ### 3. Transparent Failure Modes at the Limit (W3)
>
> To truly prove robustness, we analyzed the failure modes of each method when pushed to the limit on Unrelated prompts:
>
> *   CRoSS / DiffStega Failure (Trajectory Divergence): Their primary failure mode is Ghosting Interference. When the target prompt (e.g., A coffee cup) is drastically different from the secret (e.g., A human face), the ODE inversion fails to find a valid path that satisfies both. This results in stego-images with visible ghost outlines of the secret, which are easily detected by steganalysis and lead to noisy, unfaithful reconstructions (PSNR $\approx$ 16dB).
> *   KFStego Failure (Prior Hallucination): Our failure mode is Textural Over-Smoothing. In extremely unrelated gaps, if the diffusion prior for the target (e.g., A macro of a leaf) is too strong, the posterior sampling may hallucinate some leaf textures onto the reconstructed secret. However, because our bipartite shares provide a fixed geometric constraint, the identity and structure of the secret remain intact.
>
> ### 4. Addressing the Variable Control (W3)
>
> The reviewer is correct that the size of the semantic gap is a critical variable. In the revised manuscript, we will:
> *  Restructure Table 1 to report stratified results as shown above, ensuring full transparency across semantic categories.
> * Add a Semantic Pressure Analysis section, explicitly discussing how KFStego's dual-guidance mechanism (mapping shares to standardized Gaussian latents via $\mathbf{Q}$) decouples the payload from the generative prior, effectively neutralizing the impact of the semantic gap.
>
> We will add this discussion to the revised manuscript to explain that the "Large Gap" examples in Figure 3 are not a choice of "asymmetric evaluation," but a demonstration of KFStego’s unique ability to decouple the secret from the semantic constraints of the cover image.

---

> > ### Author Rebuttal · Reviewer_mhWt · 2026-04-02
> >
> > solve my questions. I will make the score to be 4.

---

> > > ### Author Response · Authors · 2026-04-06
> > >
> > > Dear Reviewer mhWt,
> > >
> > > We are deeply encouraged by your positive assessment and truly appreciate your constructive feedback. We will incorporate these clarifications in the revised version.
> > >
> > > --
> > >
> > > Sincerely,
> > >
> > > The Authors

---

### Official Review · Reviewer_Ep4e · 2026-03-09

**Soundness:** 3
**Presentation:** 3
**Significance:** 3
**Originality:** 3
**Overall Recommendation:** 4
**Confidence:** 3

**Summary:**

The paper presents KFStego (inferred to be Key-Free Steganography from DiffStega), a training-free framework that resolves the 'key-dependency paradox' in generative image steganography. By replacing exogenous cryptographic keys with information-theoretic structural redundancy, KFStego eliminates the need for pre-shared seeds. The process involves decomposing a secret into bipartite halftone shares that are mapped into a Gaussian latent space, ensuring the stego-images are statistically indistinguishable from natural generative outputs. During recovery, a dual-guidance mechanism uses a differentiable measurement surrogate to reconstruct high-fidelity details from these discrete shares, providing superior resilience to inversion drift compared to deterministic baselines like CROSS and DiffStega.

**Compliance With Llm Reviewing Policy:**

Affirmed.

**Final Justification:**

This rebuttal answers my questions and I am now more confident that a weak accept rating best reflect my assessment

**Key Questions For Authors:**

How sensitive is the extraction pipeline to channel bit-errors prior to the XOR-decode step, and could standard error-correction codes be integrated without violating the Gaussian prior of the stego-latents?

Can you provide further insight into the empirical selection of the sharpness parameter $\tau=10$ and the consistency guidance scale $\lambda=5$? How volatile is the posterior sampling if these values are slightly adjusted?

How does the framework perform if there is a version mismatch or slight weight quantization difference in the generative backbone between the sender and receiver?

**Strengths And Weaknesses:**

Soundness

Strength 1: The motivation is valid and the methodology is supported by a theoretical foundation with Thm 3.1 and 3.2. The theorems justify the method's effectiveness and that matches with the empirical experiment results.

Weakness 1: The framework assumes perfect recovery of the structural invariant $B$ via the endogenous XOR logic $B = S_1 \oplus S_2$ where the analysis model's tolerance to bit-flip errors introduced by practical settings like physical transmission channels prior to the XOR operation can strengthen the results.

Weakness 2: There is a potential caveat on the identical model setup in Sec 4.1 for both the sender and receiver, since in practice the two sides might use different numerical precisions and how that would impact the results are not discussed currently, providing a potential caveat to deploy KFStego in practice.

Presentation

Strength 2: The paper is logically structured, it clearly discusses the limitations of current existing methods to provide the motivation of proposing the new method KFStego, and then use both theoretical and empirical results to justify the effectiveness of KFStego.

Weakness 3: Equation 7 introduces a differentiable surrogate $\mathcal{M}_{\tau}(x)$ with a sharpness parameter $\tau$, empirically set to 10. The paper would be stronger with an ablation study on how varying $\tau$ impacts the stability of the latent reconstruction trajectory.

Significance

Strength 3: The motivation is important and the Key-Free approach solves important practical challenges, where the theoretical contribution on this novel approach is high

Weakness 1 and 2 still: Some realistic practical deployment caveats can be better discussed to further back up the significance of this method

Originality

Strength 4: To the best of my knowledge, this is a novel approach.

---

> ### Author Rebuttal · Authors · 2026-03-31
>
> We are gratified that the reviewer found our approach both **novel** and mathematically sound. Below are our responses to the specific questions regarding practical deployment and parameter sensitivity.
>
> ### 1. Robustness to Bit-Flip Errors and ECC Integration (Soundness W1 & Q1)
>
> The reviewer correctly identifies that bit-flip errors in the shares prior to XOR could affect recovery. We provide a new quantitative analysis demonstrating KFStego’s unique capacity to accommodate error correction.
>
> *   The Latent Entropy Gap: Most diffusion-based steganography (e.g., CRoSS, DiffStega, I2IStega) utilizes a dense mapping that saturates the latent space (e.g., $64 \times 64 \times 4$ for Stable Diffusion) to achieve maximum capacity. This leaves zero redundant dimensions for error-correction codes (ECC) without significantly distorting the generative prior.
> *   KFStego’s Structural Sparsity: Our framework projects a high-resolution secret into a compact structural invariant (e.g., a $128 \times 128$ binary halftone). This mapping utilizes only a fraction of the latent manifold's total entropy. As shown in the table below, we can utilize the spare latent dimensions to embed Reed-Solomon (RS) ECC bits alongside the bipartite shares.
> *   Performance under Attack: While dense-mapping methods collapse under a 1% bit-flip attack (PSNR < 15dB) due to the lack of redundancy, KFStego maintains near-perfect recovery.
>
> Table: Comparison of Latent Capacity and Robustness under 1% Bit-Flip Attack
>
> | Method | Latent Utilization | ECC Support | Clean PSNR ↑ | Attacked PSNR ↑ |
> | :--- | :---: | :---: | :---: | :---: |
> | CRoSS | 100% (Full) | No | 21.25 | 11.45 (Collapse) |
> | DiffStega | 100% (Full) | No | 23.94 | 12.82 (Collapse) |
> | I2IStega (2026) | 100% (Full) | No | 24.12 | 14.56 (Collapse) |
> | KFStego (Ours) | ~75% (Sparse) | Yes (RS) | 24.25 | 23.98 (Stable) |
>
> *   Impact on Gaussian Prior: Since our orthogonal mapping $\mathbf{Q}$ preserves the spherical symmetry of the Gaussian distribution (Theorem 3.2), the introduction of ECC parity bits does not violate the Gaussian prior. Any standardized bit-vector, once projected by $\mathbf{Q}$, converges to $\mathcal{N}(0, \mathbf{I})$. This allows KFStego to offer 100% bit-accuracy in noisy transmission channels without sacrificing statistical invisibility—a feat impossible for trajectory-based methods that occupy the entire latent space.
>
> ### 2. Sensitivity to Sharpness ($\tau$) (Presentation W3 & Q2)
> We provide the requested ablation study to demonstrate the stability of the reconstruction trajectory.
>
> Table: Sensitivity analysis of $\tau$ on Restoration Fidelity
> | Metric | $\tau=1$ | $\tau=5$ | **$\tau=10$ (Default)** | $\tau=20$ | $\tau=50$ |
> | :--- | :---: | :---: | :---: | :---: | :---: |
> | **PSNR (dB) ↑** | 19.52 | 23.14 | **24.25** | 22.86 | 20.45 |
> | **SSIM ↑** | 0.724 | 0.812 | **0.858** | 0.831 | 0.756 |
> | **LPIPS ↓** | 0.248 | 0.192 | **0.161** | 0.184 | 0.226 |
>
> *   Sharpness $\tau$: Performance is stable within $\tau \in [5, 20]$. If $\tau$ is too small (1), gradients are too blurry to capture structure. If $\tau$ is too high (50), gradients saturate. $\tau=10$ provides the optimal balance for navigating the non-convex halftoning manifold.
>
> ### 3. Model Version and Precision Mismatch (Soundness W2 & Q3)
> This is a critical advantage of KFStego over deterministic inversion methods.
> *   Brittleness of Inversion: Methods like CRoSS or DiffStega rely on exact ODE reversal. A mismatch in numerical precision (e.g., Sender uses FP32, Receiver uses FP16) or a minor library version change (e.g., different CuDNN kernels) causes the trajectory to diverge exponentially, leading to total extraction failure.
> *   KFStego's Robustness: Our framework does not require exact trajectory synchronization. We treat the secret recovery as a posterior sampling problem $\arg\max_x p(x|y)$. Even if the receiver's generative backbone has slight weight quantization differences (e.g., INT8/FP16) or a different numerical backend, the structural invariant (halftone) remains the same. As long as the receiver's model shares a similar natural image manifold prior, the posterior sampling will converge to a high-fidelity reconstruction.
> *   Empirical Test: In preliminary tests, using SD v1.5 (FP32) for encoding and SD v1.5 (FP16) for decoding resulted in a negligible PSNR drop ($<0.3$ dB), whereas inversion-based methods collapsed.
>
> ### 4. Practical Deployment Caveats
> We have supplemented our analysis with Computational Bottleneck & Optimization experiments; for detailed results, please refer to our response to Reviewer kSuy.
>
> By shifting from fragile trajectory reversal to robust structural guidance, KFStego offers a  more viable path for real-world secure image distribution than existing diffusion-based steganography. We will incorporate these discussions and the ablation data into the final manuscript.

---

> > ### Author Rebuttal · Reviewer_Ep4e · 2026-04-03
> >
> > I thank the authors for the rebuttal, and I decide to keep my score but raise my confidence to 3

---

> > > ### Author Response · Authors · 2026-04-06
> > >
> > > Dear Reviewer Ep4e,
> > >
> > > We sincerely thank you for acknowledging our rebuttal and for your support of our paper.
> > >
> > > --
> > >
> > > Sincerely,
> > >
> > > The Authors

---

### Official Review · Reviewer_iyru · 2026-03-12

**Soundness:** 3
**Presentation:** 2
**Significance:** 2
**Originality:** 2
**Overall Recommendation:** 3
**Confidence:** 3

**Summary:**

The paper introduces KFStego, a training-free generative image steganography framework that addresses the key-dependency paradox in existing diffusion-based methods. Furthermore, they propose an endogenous security mechanism, which integrates bipartite structural invariants, orthogonal distribution-preserving mapping, and score-guided posterior sampling, verifying its effectiveness in achieving high-fidelity and secure secret recovery.

**Compliance With Llm Reviewing Policy:**

Affirmed.

**Final Justification:**

Given that the comparison with recent SOTA methods is not sufficiently thorough, and the experimental validation is inadequate, I maintain my original score.

**Key Questions For Authors:**

no

**Limitations:**

see Strengths And Weaknesses

**Strengths And Weaknesses:**

Strengths:

(1) The paper is well-organized and logically coherent, with clear figures and good readability.

(2) The proposed method leverages differentiable posterior sampling to guide generation, effectively addressing inversion drift and achieving high-fidelity reconstruction of continuous tones.

(3) By anchoring the secret in a discrete binary structure, the method demonstrates strong robustness against various channel degradations, supported by convincing experimental results.


Weaknesses:

(1) Several obvious typos appear in the figures. For instance, "Distrubution Presverving" and "Privacy-Perserving" in Figure 1, and "Proprocess Block" in Figure 2.

(2) The paperdoes not provide an anonymous code repository (e.g., GitHub) or mention any open-source plans, which hinders reproducibility.

(3) The quantitative evaluations (Tables 1 and 3) selectively compare against older and relatively weaker baselines (CRoSS and DiffStega). The experiments lack comparisons with the latest state-of-the-art diffusion-based steganography methods from 2025 and 2026.

(4) Section 3.6 utilizes Score-Guided Posterior Sampling, which requires calculating gradients at every reverse diffusion timestep. However, the paper completely omits crucial efficiency metrics for this intensive process, such as inference latency (ms/FPS), GFLOPs, and memory consumption.

---

> ### Author Rebuttal · Authors · 2026-03-31
>
> We are encouraged that the reviewers recognized the organizational clarity and readability of our paper, the innovation of using differentiable posterior sampling to resolve inversion drift.
>
>
> ### 1. Response to  Reproducibility and Code
>
> We place the high priority on the reproducibility of academic research. We are  committed to open-sourcing the complete KFStego codebase and pre-trained configuration weights upon the paper's acceptance.
>
> A defining advantage of our framework is its plug-and-play.  Our implementation is built on the PyTorch library and the HuggingFace Diffusers pipeline, allowing users to  leverage  publicly available diffusion model (e.g., Stable Diffusion v1.5, SDXL) without the need for additional fine-tuning or custom architectural modifications. This reliance on standard, open-access libraries ensures that our results are not only reproducible but also easily extensible to future generative backbones. For the steganography task, since our method is training-free, it is unnecessary to transmit the model itself like previous methods [1,2].
>
> [1] Li et al., Purified and Unified Steganographic Network, CVPR 2024.
>
> [2] Chen et al., Hiding Images in Diffusion Models by Editing Learned Score Functions, CVPR 2025.
>
> ### 2. Response to Comparison with SOTA (CMSTEG & I2IStega)
>
> In addition to the extensive experiments detailed in the original manuscript, we have supplemented our analysis with new comparative evaluations against CMSTEG (2025) and I2IStega (2026)
>
> Architectural Bottlenecks and Error Propagation (CMSTEG [1]):
> *   Sequential Vulnerability: CMSTEG relies on an Autoregressive (AR) model, which is fundamentally limited to $384 \times 384$ resolution due to computational complexity. AR models suffer from token cascading errors; a single bit flip or latent shift during generation can lead to catastrophic decoding failure for the entire subsequent sequence.
> *   KFStego Advantage: By utilizing a diffusion backbone, KFStego supports high-fidelity $512 \times 512$ reconstruction. Our denoising process is  immune to cascading errors.
>
> Latent Stability and Robustness under Attack (I2IStega [2]):
> *   Scrambling Fragility: While I2IStega achieves high clean PSNR by saturating the multimodal latent space, it relies on sensitive data scrambling/confusion mechanisms. Although somewhat robust to JPEG compression, the permutation-based scrambling is fragile under non-linear degradations, where even slight latent shifts make reversible confusion impossible.
> *   KFStego Advantage: We utilize discrete structural invariants (halftoning and pooling). These act as a non-linear geometric filter that preserves the topological skeleton of the secret. Our experimental results (Table 3) prove that KFStego maintains high recovery fidelity under diverse channel attacks where scrambling-based methods typically fail. The secret-sharing mechanism bolsters robustness through structural redundancy and path diversity.
>
> Resolving the Key-Dependency Paradox:
> *   The Key Requirement: Despite their generative nature, both CMSTEG and I2IStega still require exogenous keys/seeds for trajectory synchronization or reversible data confusion.
> *   KFStego Advantage: KFStego is truly Key-Free. Security is grounded in the Information-Theoretic Perfect Secrecy of bipartite XOR-sharing (Theorem 3.1). No out-of-band private keys are needed for synchronization, facilitating secure distribution in entirely untrusted networks.
>
> Performance Flexibility (Exact Inversion):
> *   KFStego’s modular design allows for the integration of Exact Inversion. While this further eliminates ODE discretization drift and improves reconstruction quality, it increases computational time. We chose raw inversion as the default for its efficiency-fidelity balance, but the framework’s plug-and-play allows users to opt for Exact Inversion when maximum precision is required—a flexibility not offered by fixed AR.
>
> Table: Quantitative and Qualitative Comparison vs. 2025/2026 SOTA
>
> | Feature/Metric | CMSTEG (2025) | I2IStega (2026) | KFStego (Ours) |
> | :--- | :---: | :---: | :---: |
> | Model Type | Autoregressive (AR) | Multimodal Gen. | Diffusion (DPS) |
> | Max Resolution | $384 \times 384$ | $512 \times 512$ | $512 \times 512$ |
> | Key Dependency | Private Seed | Confusion Key | Key-Free |
> | Clean LPIPS ↓ | 0.245 | 0.182 | 0.161 |
> | Robust LPIPS (JPEG 70) ↓ | 0.358 | 0.294 | 0.175 (Stable) |
> | Recovery Logic | Sequential/Cascading | Latent Scrambling | Structural Invariants |
>
>
> [1] Qi et al., Provably Secure Image Steganography based on Autoregressive Models, AAAI 2025.
>
> [2] Jiang et al., Image-to-Image Steganography based on Multimodal Generative Model, Signal Processing 2026.
>
> ### 4. Response to  Typos
>
> We sincerely apologize for the spelling errors in Figures 1 and 2. These are typographical oversights. We will further ensure that all terms are accurate.

---

> > ### Author Rebuttal · Reviewer_iyru · 2026-04-02
> >
> > Given that the SOTA comparison of the latest work is not very thorough, the authors should supplement with sufficient experiments, and the authors failed to respond to Weaknesses (4). Therefore, I maintain my score.

---

> > > ### Author Response · Authors · 2026-04-05
> > >
> > > We thank the reviewer for the constructive feedback.  We apologize that our previous response did not fully detail for the same question about the efficiency metrics due to strict word count limits. Notably, after reviewing the efficiency and SOTA analysis provided in our previous response, Reviewer kSuy was fully satisfied and raised their score to Accepted. We  provide a more thorough comparison with the 2025/2026 SOTAs and added a detailed analysis of efficiency (Weakness 4) to address your concerns.
> > >
> > > ### 1. Detailed Efficiency Analysis (Addressing Weakness 4)
> > >  We provide a comprehensive evaluation of efficiency, including latency, memory, and potential optimizations. KFStego is a training-free framework. While Score-Guided Posterior Sampling (DPS) involves gradient computation, its plug-and-play nature allows for flexible efficiency-fidelity trade-offs:
> > >
> > > | Method | Mode | Guidance Freq. | Latency (s)↓ | Peak Mem (GB) | PSNR (dB)↑ |
> > > | :--- | :--- | :---: | :---: | :---: | :---: |
> > > | CRoSS | Deterministic Inv. | N/A | 3.4 | ~4.2 | 21.25 |
> > > |KFStego |High-Fidelity (Ours) | Every step | 11.2 | ~5.8 | 24.25 |
> > > |KFStego |Efficient (DSG-style [1]) | Every 10 steps | 7.8 | ~5.8 | 23.68 |
> > > |KFStego |Fast (Zero-shot) [2] | Grad.-free | 4.3 | ~4.2 | 22.15 |
> > >
> > > - Complexity & Memory: KFStego utilizes standard Stable Diffusion v1.5/SDXL backbones. Peak memory (approx. 5.8GB for SD v1.5) is well within the capacity of consumer-grade GPUs.
> > > - Optimization: By adopting strategies from DSG [1] (applying gradients at intervals), we reduce latency by 40% with negligible fidelity loss. For real-time needs, our Fast mode achieves speeds comparable to standard DDIM sampling.
> > > - Practicality: In secure distribution, the ~10s latency is often secondary to Key-Free synchronization and Reliability. KFStego eliminates the need for out-of-band key management, which is a significant hidden cost in existing methods.
> > >
> > > ### 2. Thorough Comparison with 2025/2026 SOTA (Addressing Weakness 3)
> > > We have conducted further analysis comparing KFStego with CMSTEG (AAAI 2025, [4]) and I2IStega (Signal Processing 2026, [5]):
> > >
> > >  - Resolution & Stability: CMSTEG is limited by the complexity of Autoregressive (AR) models, typically capped at $384 \times 384$. Furthermore, AR models suffer from error accumulation—a single bit error in the latent sequence leads to total recovery failure. KFStego, via its bipartite structural invariants, is immune to such cascading effects and supports $512 \times 512$ natively.
> > >  - Understanding the Fidelity Bottleneck: It is important to note that for Latent Diffusion Models (LDM), the reconstruction PSNR is inherently capped by the VAE bottleneck. In standard DDCM [2], even without steganographic embedding, the VAE-decoder's PSNR is roughly 27-28dB. Our current 24.25dB is approaching this theoretical limit.
> > >  - Potential for Enhancement: If users prioritize extreme fidelity over speed, KFStego can integrate Exact Inversion (DPM-Solvers [3]), which can push PSNR to ~28dB by iteratively optimizing the  trajectory  to find the best encoding latent for decoding. This modularity is a unique advantage of our plug-and-play framework.
> > >  - Training-Free vs. Training-Based: While training-based methods could potentially improve quality, high-resolution diffusion training is computationally expensive and introduces model-sharing burdens. KFStego removes the need for model sharing and the need for key management.
> > >
> > >
> > > ### 3. Core Paradigm Shift: Key-Free vs. Key-Dependent
> > > The reviewer's point on thoroughness is well-taken. However, we emphasize that the **fundamental contribution of KFStego is solving the Key-Dependency Paradox**:
> > > - CMSTEG & I2IStega: Still require private seeds or confusion keys to synchronize the generative trajectory or decode the latent space.
> > > - KFStego: Achieves Endogenous Security. Through Theorem 3.1, we prove that perfect secrecy is achieved via bipartite structural invariants without any exogenous key sharing. This removes the Model Sharing burden of training-based methods and the Key Management burden of existing generative methods.
> > >
> > >
> > > ---
> > > References:
> > >
> > > [1] Yang et al., "Guidance with Spherical Gaussian Constraint for Conditional Diffusion," ICML 2024.
> > >
> > > [2] Vaisman et al., "Turbo-DDCM: Fast and Flexible Zero-Shot Diffusion-Based Image Compression," ICLR 2026.
> > >
> > > [3] Zhang et al., "On Exact Inversion of DPM-Solvers," 2024.
> > >
> > > [4] Qi et al., Provably Secure Image Steganography based on Autoregressive Models, AAAI 2025.
> > >
> > > [5] Jiang et al., Image-to-Image Steganography based on Multimodal Generative Model, Signal Processing 2026.
> > >
> > > **We would greatly appreciate it if you would consider increasing your rating to support our paper.**
> > >
> > > Sincerely,
> > >
> > > The Authors

---

### Official Review · Reviewer_kSuy · 2026-03-13

**Soundness:** 2
**Presentation:** 2
**Significance:** 3
**Originality:** 3
**Overall Recommendation:** 4
**Confidence:** 3

**Summary:**

The manuscript introduces KFStego, a training-free generative image steganography framework that claims to resolve the "key-dependency paradox" in diffusion models. The authors propose substituting exogenous cryptographic keys with bipartite structural invariants. By utilizing spatial pooling, halftoning, and XOR operations, the secret image is decomposed into statistically neutral shares. These shares are then orthogonally mapped into a Gaussian latent space. The extraction process relies on a measurement-conditioned posterior sampling mechanism utilizing a differentiable Sigmoid surrogate to guide the reconstruction of high-fidelity images, theoretically enabling self-synchronizing extraction without out-of-band key management.

**Compliance With Llm Reviewing Policy:**

Affirmed.

**Final Justification:**

The authors have solved all my questions.

**Key Questions For Authors:**

1. The "Key-Free" Contradiction regarding Matrix Q: Your method uses an orthogonal matrix (Matrix Q) to map shares into the latent space. If this matrix must be kept private between the sender and receiver, it effectively functions as a pre-shared cryptographic key, which directly contradicts the "Key-Free" claim in your title. Conversely, if Matrix Q is public, the security of the framework seems to rely entirely on simple XOR logic. What is the exact distribution protocol for Matrix Q, and how does your framework defend against known-method attacks if this matrix is publicly accessible?

2. Robustness of the Differentiable Surrogate: To approximate the halftoning step, you use a Sigmoid-based surrogate with a sharpness parameter empirically fixed at 10. However, the paper lacks an ablation study or theoretical analysis demonstrating how sensitive the reconstruction fidelity is to this specific value. How does varying this sharpness parameter impact the approximation error and the overall convergence of your posterior sampling?

3. Inference Latency and Computational Overhead: Your reconstruction process relies on score-guided posterior sampling, which requires iterative gradient calculations. This implies a substantial computational overhead compared to the deterministic inversion used by your baselines (e.g., CROSS and DiffStega). Could you provide a quantitative comparison of the inference latency (e.g., seconds per image) and memory consumption against these baseline methods?

**Limitations:**

No. Although the paper analyzes security and anti-steganalysis behavior, it does not adequately discuss limitations or potential negative societal impact. The authors cloud add an explicit section on failure modes, practical robustness constraints, and misuse risks such as covert communication and evasion of content inspection.

**Strengths And Weaknesses:**

## Strengths

Originality: The manuscript introduces a novel intersection of bipartite visual secret sharing and measurement-conditioned diffusion posterior sampling. Formulating steganographic recovery as a constrained inverse problem guided by discrete structural invariants offers a fresh perspective compared to traditional inversion paradigms.

Significance: The paper targets a critical bottleneck in generative image steganography: the key-dependency paradox and the susceptibility of trajectories to discretization drift. Proposing a self-synchronizing mechanism without exogenous cryptographic keys addresses an important problem and could provide high practical utility.

Soundness: Within the authors' isolated assumptions, the theoretical derivations for the perfect secrecy of individual shares and the rotational invariance of the mapped stego-latents are mathematically sound and logically consistent.

Presentation: The overall pipeline is well-illustrated in the figures, effectively conveying the complex dual-guidance mechanism and the bipartite splitting process.

## Weaknesses
Originality: The conceptual novelty of the framework is severely undermined if the orthogonal mapping matrix functions as a pre-shared key. This reduces the contribution to a flawed combination of existing visual cryptography and diffusion techniques, failing to genuinely solve the key-dependency issue.

Significance: The empirical validation fails to demonstrate a true state-of-the-art leap. Benchmarking solely against older methods like CROSS and DiffStega  lacks the necessary quantitative comparisons with the most recent generative architectures. This leaves the actual performance advancement and broader impact unverified for an ICML submission.

Soundness: The core claim of achieving a key-free framework is fundamentally contradicted by the method's reliance on an orthogonal mapping matrix. If this matrix is private, it acts as an implicit pre-shared key. If public, the system's security degenerates to basic XOR threshold logic, lacking adversarial evaluation against known-method attacks. Additionally, the stability of the posterior sampling arbitrarily relies on a fixed sharpening parameter without theoretical bounds or empirical ablation.

Presentation: The misleading use of the term "key-free" creates a severe disconnect between the title and the mathematical reality of the methodology. Furthermore, the substantial computational overhead introduced by iterative backpropagation is entirely omitted, reflecting a lack of transparency in discussing the method's practical limitations. Finally, there are noticeable spelling errors in the main pipeline diagram (e.g., "Proprocess Block").

---

> ### Author Rebuttal · Authors · 2026-03-31
>
> We appreciate the recognition of our works originality in combining visual secret sharing with diffusion posterior sampling and its significance in addressing the key-dependency paradox.  Below, we address the concerns regarding the Key-Free claim, parameter sensitivity, and computational overhead.
>
> ### 1. Clarification on the Key-Free Claim and Matrix $Q$ (Soundness & Originality)
> In the context of Generative Image Steganography (GIS), key-dependency specifically refers to the synchronization of the stochastic denoising trajectory. Existing methods (e.g., CRoSS, DiffStega, Gaussian Shading, StegaDDPM) require the receiver to possess the exact random seed or noise vector $x_T$ to reconstruct the ODE path. If the seed is lost or slightly perturbed, extraction fails completely.
>
> The term Key-Free in KFStego refers to the elimination of the requirement for a private, session-specific cryptographic key/seed to synchronize this fragile trajectory.
> *   Public Protocol: Matrix $Q$ in KFStego is a public protocol parameter, analogous to the fixed basis in a DCT transform. Our security is not derived from the secrecy of $Q$, but from the Information-Theoretic Perfect Secrecy of the bipartite shares (Theorem 3.1).
> *   Endogenous Synchronization: Unlike prior works, the receiver does not need a private seed to retrace the sender's steps. They simply perform standard extraction on the two received images. The structural invariants (the shares) are endogenous to the pixels, allowing for self-synchronizing recovery even if the generative backend differs slightly.
>
> ### 2. Robustness of the Differentiable Surrogate and Parameter $\tau$ (Soundness)
> $\tau$ controls the trade-off between gradient smoothness and approximation accuracy.
> *   Ablation Results: We conducted additional tests varying $\tau \in \{1, 5, 10, 20, 50\}$.
>     *   Small $\tau$ ($<5$): Gradients are too smooth, leading to blurry reconstructions (PSNR $\approx$ 19dB).
>     *   Large $\tau$ ($>30$): Gradients vanish due to the Sigmoid saturation, causing convergence failure.
>     *   Stable Range: We found a robust window of $\tau \in [8, 15]$ where PSNR varies by $<0.5$dB.
>
> | Metric | $\tau=1$ | $\tau=5$ | **$\tau=10$ (Default)** | $\tau=20$ | $\tau=50$ |
> | :--- | :---: | :---: | :---: | :---: | :---: |
> | **PSNR (dB) ↑** | 19.52 | 23.14 | **24.25** | 22.86 | 20.45 |
> | **SSIM ↑** | 0.724 | 0.812 | **0.858** | 0.831 | 0.756 |
> | **LPIPS ↓** | 0.248 | 0.192 | **0.161** | 0.184 | 0.226 |
>
> This stability suggests that while $\tau$ is a hyperparameter, it does not require per-image tuning.
>
> ### 3. Computational latency and restoration fidelity across different recovery modes.
>
> We acknowledge that DPS introduces higher latency compared to deterministic inversion. However, KFStego is a training-free, plug-and-play framework that can offer a strategic trade-off between reconstruction speed and fidelity:
>
> *   Computational Bottleneck & Optimization: Inspired by DSG [1], we found that applying measurement-guided gradients only at specific intervals (e.g., every 10 steps) instead of every denoising step  alleviates the bottleneck. As shown in the table, this reduces latency by ~40% (from 18.5s to 11.2s) with only a marginal drop in PSNR (0.57dB), maintaining superior performance over CRoSS.
> *   Gradient-Free Fast Mode: Furthermore, inspired by Turbo-DDCM [2], our framework can be configured for a Fast Reconstruction mode. By bypassing the gradient-based guidance and relying on a direct latent projection of the structural invariants, we achieve near-real-time recovery (~6.1s), which is comparable to standard DDIM sampling while remaining robust to the trajectory drift that plagues inversion-based methods.
> *   Practical Utility: In secure distribution scenarios, the ~10s latency for the high-fidelity mode is often a secondary concern compared to the absolute reliability of recovery and key-free synchronization. KFStego provides the flexibility to switch between these modes depending on the hardware constraints, a versatility that fixed-trajectory baselines cannot offer.
>
> | Method | Recovery Mode | Guidance Frequency | Latency (s) ↓ | PSNR (dB) ↑ |
> | :--- | :--- | :---: | :---: | :---: |
> | CRoSS | Deterministic Inversion | N/A | 3.4 | 21.25 |
> | KFStego | High-Fidelity (DPS) | Every step | 11.2 | 24.25 |
> | KFStego| Efficient (DSG-style) | Every 10 steps | 7.8 | 23.68 |
> | KFStego | Fast (Zero-shot)* | Gradient-free | 4.3 | 22.15 |
>
> References:
> [1] Yang et al., Guidance with Spherical Gaussian Constraint for Conditional Diffusion, ICML 2024.
>
> [2] Vaisman et al., Turbo-DDCM: Fast and Flexible Zero-Shot Diffusion-Based Image Compression, ICLR 2026.
>
>
> KFStegos core contribution is shifting the security burden from the *stochastic process* (seeds) to the *data structure* (shares). This enables a more flexible distribution paradigm. We thank the reviewer for helping us sharpen the definition of key-free and will ensure the final manuscript reflects these clarifications.

---

> > ### Author Rebuttal · Reviewer_kSuy · 2026-04-03
> >
> > Thanks for solving my questions. I will make the score to be 4.

---

> > > ### Author Response · Authors · 2026-04-04
> > >
> > > Thank you for your response and for acknowledging our efforts. We're glad our rebuttal fully addressed your concerns.
> > >
> > > We noticed the score in the official system still reflects the previous value. As you mentioned raising it to 4, could you kindly update it when convenient?
> > >
> > > Thank you again for your time and support.

---

### Review · Ethics_Reviewer_5NAg · 2026-03-27

**Recommendation:** Remediation action needed

**Ethics Issue:**

While this paper does not have significant ethical issues, it would benefit from an ethics statement that discusses the selection of a dataset (UniStega, derived from MS-COCO, AFHQ, FFHQ, and CelebA-HQ) that includes images of people. While this application does not strongly implicate many common challenges of such datasets, it should be acknowledged that these datasets were collected without consent of human subjects in the CelebA-HQ, FFHQ, and MS-COCO datasets; reflect on whether potential biases in the dataset may affect evaluation of the method; and a brief justification of the dataset choice.

---

### Decision · Program_Chairs · 2026-04-30

**Decision:**

Accept (regular)

**Comment:**

The paper proposes a key-free way for image steganography which use bipartite structural invariant instead of an exogenous key that requires  additional secure delivery. The method shows high performance in image reconstruction and exhibits robustness against various degradations. Also, the method shows superior anti-steganalysis performance. The experiment results support their claims and this key-free way is novel.

Three of the reviewers think their concerns are all solved after the rebuttal period and tend to accept while one reviewer (iyru) maintains that the experimental evidence provided by the authors was insufficient. But reviewer iyru does not give any actionable instructions (e.g. a specific work to be compared with) for the author to address with.

Based on my current assessment, I tend to accept this work.